# Completion of the Continuum of Maternity Care in the Emerging Regions of Ethiopia: Analysis of the 2019 Demographic and Health Survey

**DOI:** 10.3390/ijerph20136320

**Published:** 2023-07-07

**Authors:** Abdulaziz Mohammed Hussen, Ibrahim Mohammed Ibrahim, Binyam Tilahun, Özge Tunçalp, Diederick E. Grobbee, Joyce L. Browne

**Affiliations:** 1Julius Global Health, Julius Centre for Health Sciences and Primary Care University Medical Centre Utrecht, Utrecht University, 3584 CX Utrecht, The Netherlands; d.e.grobbee@umcutrecht.nl (D.E.G.); j.l.browne@umcutrecht.nl (J.L.B.); 2Department of Midwifery, College of Medicine and Health Science, Samara University, Samara P.O. Box 132, Ethiopia; ibrahimmaca52@gmail.com; 3Department of Health Informatics, Institute of Public Health, College of Medicine and Health Sciences, University of Gondar, Gondar P.O. Box 196, Ethiopia; binyam.tilahun@uog.edu.et; 4eHealth Lab Ethiopia, Institute of Public Health, College of Medicine and Health Sciences, University of Gondar, Gondar P.O. Box 196, Ethiopia; 5UNDP—UNFPA—UNICEF—WHO—World Bank Special Programme of Research, Development and Research Training in Human Reproduction (HRP), Department of Sexual and Reproductive Health and Research, World Health Organization, 1211 Geneva, Switzerland; tuncalpo@who.int

**Keywords:** continuum of maternity care, maternal healthcare, emerging regions, Ethiopia

## Abstract

Maternal mortality in Ethiopia was estimated to be 267 per 100,000 live births in 2020. A significant number of maternal deaths occur in the emerging regions of the country: Afar, Somali, Gambela, and Benishangul-Gumuz. Achieving the Sustainable Development Goal (SDG) target requires a dramatic increase in maternal healthcare utilisation during pregnancy, childbirth, and the postpartum period. Yet, there is a paucity of evidence on the continuum of maternity care utilisation in Ethiopia, particularly in the emerging regions. Therefore, this study aimed to assess completion and factors associated with the continuum of maternity care in the emerging regions of Ethiopia. This study used the 2019 Ethiopian Demographic and Health Survey data (n = 1431). Bivariable and multivariable logistic regression analyses were carried out to identify factors associated with the completion of the continuum of maternity care. An adjustment was made to the survey design (weight, stratification, and clustering). 9.5% (95% Confidence Interval (CI): 7.0–13.0) of women completed the continuum of maternity care (four or more antenatal care, institutional delivery, and postnatal care within 24 h). Living in Somali (adjusted Odds Ratio (aOR): 0.23, 95%CI: 0.07–0.78) and Benishangul-Gumuz (aOR 3.41, 95%CI: 1.65–7.04) regions, having a secondary and higher educational level (aOR 2.12, 95%CI: 1.13–4.00), and being in the richest wealth quintile (aOR 4.55, 95%CI: 2.04–10.15) were factors associated with completion of the continuum of maternity care. Although nearly half of the women had one antenatal care, fewer than 10% completed the continuum of maternity care. This indicates that women in these regions are not getting the maximum health benefits from maternal healthcare services, and this might contribute to the high maternal death in the regions. Moreover, the completion of the continuum of maternity care was skewed toward women who are more educated (secondary or higher education) and in the richest quintile.

## 1. Introduction

Although remarkable progress has been made in the last two decades, maternal mortality is still unacceptably high in the world. In 2020, 287,000 women died due to causes related to pregnancy and childbirth—about 800 women every day [1]. Sub-Saharan Africa accounts for 70% of the deaths, and the lifetime risk of maternal death in the region is 1 in every 40 women [1]. Postpartum haemorrhage, hypertensive disorders of pregnancy, sepsis, and abortion are the leading direct causes of maternal mortality [2]. The world countries agreed to reduce the global Maternal Mortality Ratio (MMR) to less than 70 per 100,000 live births by 2030, and no country should have maternal mortality more than double the global target [3]. Maternal mortality in Ethiopia is estimated to be 267 per 100,000 live births in 2020 [1]. A significant number of maternal deaths occur in the emerging regions of the country, which are Afar, Somali, Gambela, and Benishangul-Gumuz [4,5,6].

These regions are the least developed regions in the country, characterised by harsh weather conditions, poor infrastructure, poor access to health services, low administrative capacity, and a high level of poverty [7,8]. Moreover, the scattered settlement of the community is a challenge for service delivery [7,9]. Achieving the Sustainable Development Goal (SDG) target, of less than 140 maternal deaths per 100,000 live births, requires a dramatic increase in maternal healthcare utilisation during pregnancy, childbirth, and the postpartum period, and improving the quality of care delivered throughout the country, particularly in emerging regions [10,11]. Antenatal, delivery and postnatal care are key health sector interventions for maternal survival [12,13]. Ensuring a continuum of maternity care across these three services has become a rallying call to reduce maternal and neonatal deaths [14]. Continuous uptake of antenatal, delivery, and postnatal care is associated with reduced risk of obstetrics complications and adverse birth outcomes [15,16,17].

Previous studies conducted in the emerging regions assessed the uptake of maternal healthcare and associated factors in a particular stage, pregnancy, delivery, or postpartum, which does not assure that all women receive a package of interventions starting pregnancy to the postpartum period by assessing each maternal service separately [18,19,20,21,22,23,24,25,26,27,28]. Detailed information on the completion of the continuum of maternity care in these regions would be helpful for regional and national decision-makers, and programme managers working on improving maternal health. Moreover, understanding the contributing factors to the completion of the continuum of maternity care helps in planning programmes, priority setting, and allocating resources. Therefore, this study aimed to assess completion and factors associated with the continuum of maternity care in the emerging regions of Ethiopia.

## 2. Materials and Methods

### 2.1. Data Source

The study was conducted using the 2019 Ethiopian Demographic and Health Survey (EDHS) data. During the survey a two-stage stratified cluster sampling approach was used. Each region (nine regions and two administrative cities) was stratified into urban and rural areas. In the first stage, Enumeration Areas (EAs) were chosen with probability proportional to EA size. In stage two, following a household listing in all selected EAs, a fixed number of households were selected from each EA. To gather all relevant population health information, various data collection tools were utilised, including the woman’s questionnaire with questions about maternal and child heath [29].

### 2.2. Population

Women aged 15–49 who had a live birth in the five years before the survey, and are residents of the emerging regions of Ethiopia (Afar, Somali, Gambela, or Benishangul-Gumuz), were included. Figure 1 shows the sample size and the number of women included in the survey.

### 2.3. Study Variables

Completion of the continuum of maternity care was the outcome variable for this study. It was constructed into a binary variable with complete coded as 1 and incomplete coded as 0. Continuum of maternity care was assumed completed if a woman had at least four Antenatal Care (ANC) contacts, gave birth in a health institution, and received a postnatal check within 24 h after delivery during the most recent pregnancy.

Existing published literature on the completion of the continuum of maternity care in Ethiopia were reviewed [30,31,32,33,34,35,36,37,38]. Variables that had a significant association with the completion of the continuum of maternity care or components of maternity care in the previous studies and are available in the 2019 EDHS data set were included as independent variables in the analysis. These variables were maternal age at the time of delivery, residence, marital status, highest level of maternal education, sex of the head of the household, household wealth index, number of children, and mode of delivery. The variables were catagorised based on previous studies and the distribution of responses in the data.

### 2.4. Data Analysis

First, the data were checked for completeness and cleaned. The outcome variable, the completion of the continuum of maternity care, was computed, and covariates were categorised and coded. Tables and figures were used to present the descriptive summary of the data. Bivariable and multivariable analysis was performed to identify factors associated with the completion of the continuum of maternity care.

Crude Odds Ratio (OR) and adjused Odds Ratio (aOR) with a 95% Confidence Interval (CI) were calculated to measure the strength of association between the dependent and independent variables. The association between each independent variable with the dependent variable was tested using simple logistic regression. Variables with a P value less than 0.25 in the bivariable logistic regression were selected and fitted into a multivariable logistic regression model to identify independently associated factors by overcoming the effect of confounding variables. Variance Inflation Factor (VIF) was used to test the presence of collinearity between the independent variables. The result showed that no evidence of multi-collinearity among the independent variables remained in the final model.

In the final model, associations were deemed significant when the *p*-value is less than 0.05. An adjustment was made to the survey design (weight, stratification, and clustering). We followed the same procedure to identify factors associated with utilisation of components of maternity care (ANC4+, institutional delivery, and postnatal care within 24 h). The analyses were carried out using R research software version 4.0.3 (The R foundation for statistical computing, Vienna, Austria).

## 3. Results

### 3.1. Characteristics of the Study Participants

Out of 1439 women who had a live birth in the five years before the survey, data from 1431 of them were included in the analysis; the remaining eight observations were excluded due to incomplete information. The mean age of the participants was 26.6 years (SD = 6.6 years), most of them had no education (72.7%), more than one-third (37.7%) live in a female-headed household, and more than half of them (63.0%) live in the poorest wealth quintile. The characteristics of the study participants are shown in Table 1.

### 3.2. Maternal Healthcare Utilisation

Nearly half (44.6%, 95% CI, 38.7–51.0%) of the study participants had at least one ANC visit, however, fewer than a quarter (21.6%, 95% CI, 17.6–26.0) of them had four or more ANC visits during their recent pregnancy. Over one-third (35.1%, 95% CI, 27.7–43.0) gave birth in health institutions, and less than one-fifth (19.7%, 95% CI, 16.1–24.0) of women received a Postnatal Care (PNC) within 24 h after delivery (Figure 2).

### 3.3. Continuum of Maternity Care

Out of the 21.6% (95% CI, 17.6–26.0) women who had four or more ANC visits, 15.6% (95% CI, 12.2–20.0) gave birth in health institutions, and 9.5% (95% CI: 7.0–13.0) of them received postnatal care within 24 h after the institutional delivery (Figure 3).

### 3.4. Factors Associated with Maternal Health Care Utilisation

#### 3.4.1. ANC4+

The crude OR tables for factors associated with the three maternal health care services, ANC4+, institutional delivery, and PNC within 24 h, are attached as a supplement (Appendix A). Women who live in urban areas were more likely to have four or more ANC visits than their rural counterparts [aOR 3.59 (95% CI: 1.37–9.46)]. Different results were seen across regions: women who live in Benshangul-Gumuz [aOR 2.00 (95% CI: 1.11–3.60)] were more likely to have four or more ANC visits, and those who live in Somali [aOR 0.26 (95% CI: 0.13–0.54)] and Gambela [aOR 0.41 (95% CI: 0.21–0.79)] were less likely to receive four or more ANC visits compared to women who live in the Afar region. Moreover, women who had primary education [aOR 2.11 (95% CI: 1.31–3.40)], and secondary and higher education [aOR 2.91 (95% CI: 1.60–5.31)] were more likely to have four or more ANC visits than those who had no education.

#### 3.4.2. Institutional Delivery

Women who live in urban areas of the regions [aOR 3.17 (95% CI: 1.23–8.18)] were more likely to give birth in health institutions than their rural counterparts. The odds of institutional delivery were higher for women living in Benishangul-Gumuz [aOR 4.70 (95% CI: 1.95–11.32)] and Gambela [aOR 2.61 (95% CI: 1.20–5.69)] compared to women living in Afar. Women who live in female-headed households [aOR 1.96 (95% CI: 1.26–3.05)] were more likely to give birth in health institutions. Women who are in the poorer [aOR 2.24 (95% CI: 1.34–3.75)], richer [aOR 3.42 (95% CI: 1.52–7.68)], and richest [aOR 6.65 (95% CI: 3.00–14.73)] wealth quintiles were more likely to give birth in health institutions compared to women in the poorest wealth quantile. In addition, women who had three to four children [aOR 0.43 (95% CI: 0.22–0.86)] were less likely to give birth in health institutions compared to those who had fewer.

#### 3.4.3. PNC within 24 h

Compared to women living in the Afar region, the odds of receiving PNC within 24 h were higher for women living in the Benishangul-Gumuz region [aOR 2.23 (95% CI: 1.16–4.28)] and lower in the Somali region [aOR 0.43 (95% CI: 0.24–0.78)]. Women who had secondary or higher education [aOR 4.80 (95% CI: 2.66–8.65)] were more likely to receive PNC within 24 h compared to those who had no education. The odds of receiving PNC were higher for women in the highest wealth quintile [aOR 4.55 (95% CI: 1.78–11.57)] compared to women in the poorest wealth quintile. Women who have five or more children [aOR 2.26 (95% CI: 1.28–3.99)] were more likely to receive postnatal care within 24 h compared to those who had one or two children. Moreover, women who gave birth by caesarean section [aOR 5.48 (95% CI: 1.16–26.00)] were more likely to receive a postnatal check within 24 h compared to those who gave birth vaginally (Table 2).

### 3.5. Factors Associated with Completion of the Continuum of Maternity Care

In the bivariable analysis, maternal age at the time of delivery, residence, region, maternal education, sex of the head of the household, household wealth index, and number of children were variables that had a significant association with the completion of the continuum of maternity care (*p*-value < 0.05). However, after adjusting for other variables (multivariable analysis), only region, maternal education, and household wealth index showed a significant association with the completion of the continuum of maternity care.

Compared to women living in the Afar region, the odds of completing the continuum of maternity care were higher for women living in Benishangul-Gumuz [aOR 3.41 (95% CI: 1.65–7.04)] and lower in Somali [aOR, 0.23 (95% CI: 0.07–0.78)]. Women who had secondary or higher education [aOR 2.12 (95% CI: 1.13–4.00)] were more likely to complete the continuum of care compared to women who had no education. Furthermore, the odds of completing the continuum of care were higher for women in the richest wealth quintile [aOR 4.55 (95% CI: 2.04–10.15)] compared to women in the poorest wealth quintile (Table 3).

## 4. Discussion

The study revealed that the percentage of women who had at least four ANC visits, gave birth in health institutions, and received a postnatal check within 24 h were 21.6%, 35.1%, and 19.7%, respectively. Maternal health care uptake decreases as they progress from ANC4+ to PNC. Out of the women who had four or more ANC visits, over 70% of them gave birth in health institutions. However, only less than half of these women received a postnatal check within 24 h after delivery. This finding is in line with the findings from other regions of the country [31,37,39,40] and other Sub-Saharan countries [41]. Further studies on reasons for the dropout of the continuum of maternity care should be conducted, and interventions to retain women in the continuum of maternity care should be designed and implemented accordingly.

The completion of the continuum of maternity care (ANC4+, institutional delivery, and PNC within 24 h) in the emerging regions was 9.5% (95% CI: 7.0–13.0). The absence of a single standard definition for the completion of the continuum of maternity care was a challenge to compare this finding with other studies. The definition of completion of the continuum of maternity care differs across studies. For instance, some studies assumed completion of maternity care if a woman had at least one ANC visit, skilled delivery, and a PNC check within 42 days of delivery [30,42], whereas others defined it as having ANC4+, institutional delivery, and a PNC check within 48 h postpartum [31,37]. There are also other definitions of completion of the continuum of maternity care [36,40,43,44]. Our definition is based on national and international recommendations. Although Ethiopia recently adopted and implemented the World Health Organization’s (WHO) model of eight ANC contacts [45,46], focused antenatal care (ANC4+) was the standard of care during the five years before the survey [47]. All pregnant women in Ethiopia are encouraged and supported to deliver in health facilities with skilled attendants. Regarding postnatal care, both WHO and the Ministry of Health of Ethiopia recommend that the first postnatal check should be within 24 h [48,49]. Three additional contacts are also recommended up to six weeks of the postnatal period [48,49], however, the DHS survey only captures information about the first postnatal contact. Considering these recommendations, our result shows very low percentage of completion of the continuum of maternity care, only less than 10% of women are receiving the full package of maternity care in these regions.

The most common maternal healthcare service used was ANC 1, and the percentage of women who received four or more ANC visits was lower than half of the proportion for ANC 1 visits. Region was a significant predictor of maternal health care utilisation, women who live in Benishangul-Gumuz were more likely to receive four or more ANC visits, and those who live in Somali and Gambela were less likely to receive four or more ANC visits than women who live in Afar region. The odds of institutional delivery were higher for women who live in Benishangul-Gumuz and Gambela compared to women who live in Afar. Regarding postnatal care, compared to women who live in Afar, women who live in the Benishangul-Gumuz region were more likely to receive a postnatal check within 24 h, whereas women who live in Somali were less likely to receive a postnatal check within 24 h. Furthermore, women who live in Benishangul-Gumuz were more likely to complete the continuum of maternity care and women who live in Somali were less likely to complete the continuum of maternity care than women in the Afar region.

Urban women had higher odds of receiving four or more ANC visits and were more likely to give birth in health institutions compared to their rural counterparts. This result is similar to studies conducted in Amhara, Oromia, Tigray, and Southern Nations Nationalities and People’s (SNNP) regions, and studies conducted using nationwide data [37,40,50,51,52,53]. The possible explanation could be that women in urban areas have better access to health education messages transmitted through different media outlets which could lead them to have knowledge about the benefit of receiving maternal health care. Moreover, women who live in urban areas have better access to health services compared to their rural counterparts [54].

Education showed a positive association with ANC4+ and PNC utilisation, and completion of the continuum of care. Women who had primary education, and secondary and higher education were more likely to have four or more ANC visits than those who had no education. The odds of postnatal care utilisation were higher for women who had secondary or higher education compared to those who had no education. Furthermore, women who had secondary or higher education were more likely to complete the continuum of care compared to women who had no education. This result is in line with other studies conducted in other regions of Ethiopia [30,33,36,37,40,50,55,56]. Education is likely to increase women’s health-seeking behaviour [57].

Women who live in female-headed households were more likely to give birth in health institutions compared to those who live in male-headed households. This finding is similar to studies conducted in Ethiopia, Nigeria, and India [58,59,60]. Being the head of the household might give them control over household resources and the freedom to decide on their own health needs.

Household economic status was associated with institutional delivery, PNC utilisation, and completion of the continuum of maternity care. Compared to women in the poorest wealth quintile, women in the poorer, richer, and richest wealth quintiles were more likely to give birth in health institutions. Moreover, the odds of receiving a postnatal check within 24 h, and completing the continuum of maternity care, were higher for women in the richest wealth quintile compared to women in the poorest wealth quintile. Other studies also show that maternal healthcare utilisation in Ethiopia is inequitably distributed, skewed against women in the poorest wealth quintile [61,62]. The reason could be that women in the poorest wealth quintile might not be able to cover the indirect costs of the services (transport cost and opportunity cost for the time the woman spends in a health facility), although maternal healthcare services are available for free.

Women who had three to four children were less likely to give birth in health institutions compared to those who had fewer. A similar inverse relationship was seen in studies conducted in Tanzania, India, and Nepal [63,64,65]. Due to their repeated experience, these women might perceive themselves as less risky to develop complications during delivery. In addition, the odds of receiving PNC within 24 h were higher for women who had five or more children compared to those who had one to two children. This result was in line with the finding from a nationwide study in Ethiopia [66]. Women who have five or more children are at an advanced age and have a higher risk of developing pregnancy, delivery, and postnatal complications [67,68]. Due to this reason, health professionals might give more attention to these women than to younger ones.

Furthermore, the mode of delivery was found to be associated with PNC utilisation. Women who gave birth by caesarean section were more likely to receive a postnatal check within 24 h compared to those who gave birth vaginally. This result is in line with studies conducted in other regions of Ethiopia [55,66,69,70]. Caesarean section delivery increases the risk of developing short- and long-term complications compared to vaginal delivery, particularly in low-resource settings [71]. This reason might lead health professionals to provide an immediate postnatal check to these women.

### Strengths and Limitations

The study used representative data which were collected after a rigorous sampling procedure, which makes the result generalisable to the emerging regions. Although four postnatal contacts are recommended, we only considered the first contact for our definition of the completion of the continuum of maternity care due to the absence of information about the subsequent contacts in the DHS survey. To identify the factors associated with the completion of the continuum of maternity care, only those variables that were included during the 2019 EDHS were considered. Compared to the full EDHS surveys, the 2019 EDHS (mini-EDHS) includes a smaller number of variables. Thus, it was not possible to test other potential factors. Furthermore, since women who had a live birth in the five years before the survey were asked to recall their pregnancy, delivery, and postnatal experiences, we acknowledge that there is a possibility of recall bias.

## 5. Conclusions

Although nearly half of the women had one ANC visit, fewer than 10% completed the continuum of maternity care. This indicates that women in these regions are not receiving the maximum health benefits from maternal healthcare services, and this might contribute to the high maternal death in the regions. Maternal health care utilisation decreases as they progress from ANC4+ to PNC utilisation. Health professionals should use the ANC visits as an opportunity to advise women on the importance of institutional delivery and postnatal care. Nearly 40% of women who delivered in health institutions did not receive PNC according to the national recommendation (within 24 h). Healthcare administrators should monitor the implementation of national recommendations. Moreover, the completion of the continuum of maternity care was skewed towards women who are educated (secondary and higher education) and in the richest wealth quintile. The government and partners should design and implement strategies to improve maternal healthcare utilisation among women who are poor and less educated or not educated.

## Figures and Tables

**Figure 1 ijerph-20-06320-f001:**
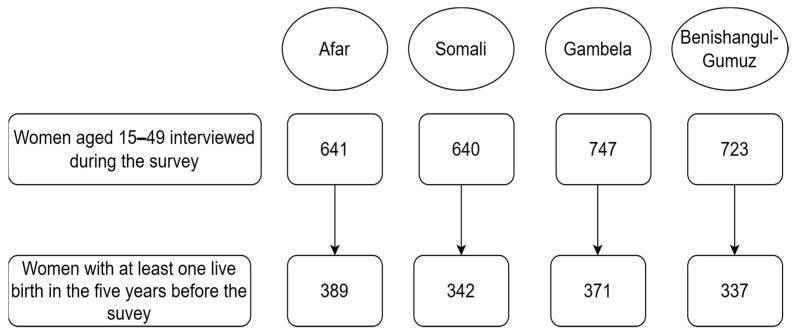
Schematic presentation of the study population and final sample size.

**Figure 2 ijerph-20-06320-f002:**
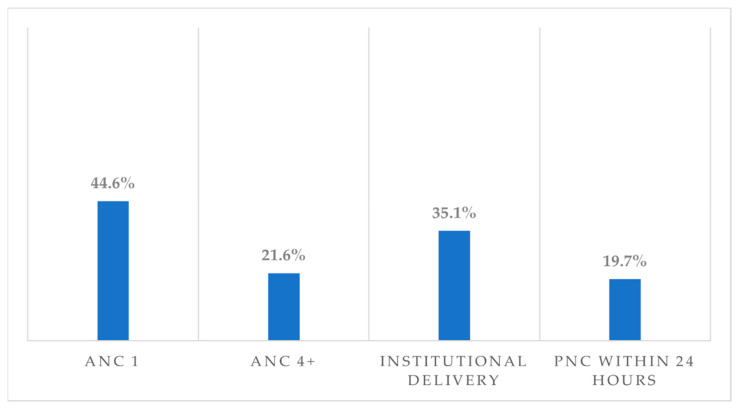
Maternal healthcare utilisation in the emerging regions of Ethiopia.

**Figure 3 ijerph-20-06320-f003:**
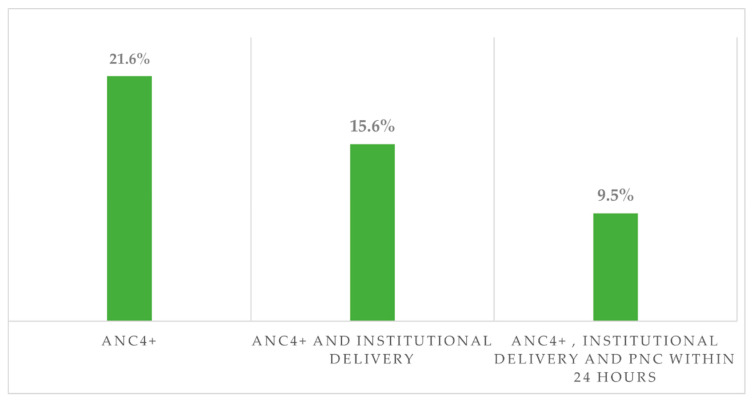
Continuum of maternity care in the emerging regions of Ethiopia.

**Table 1 ijerph-20-06320-t001:** Characteristics of the study participants (n = 1431).

Variables	Categories	Completion of the Continuum of Maternity Care			
		Yes		No	
		n	% *	n	% *
Maternal age at the time of delivery (years)	<20	39	2.0	211	11.5
	20–34	134	6.5	862	65.0
	35–49	25	1.0	160	14.0
Residence	Rural	132	4.1	1043	64.3
	Urban	66	5.5	190	26.2
Region	Afar	43	2.0	344	13.2
	Somali	8	1.9	333	63.3
	Benishangul-Gumuz	103	4.6	266	9.4
	Gambela	44	1.0	290	4.6
Marital status	Not married/not living with partner	23	1.0	102	5.7
	Married/living with partner	175	8.5	1131	84.8
Religion	Orthodox	35	1.9	146	4.8
	Catholic	0	0.0	17	0.2
	Protestant	38	1.1	222	4.5
	Muslim	124	6.5	826	80.6
	Other	1	0.0 **	22	0.5
Highest level of maternal education	No education	77	3.7	777	69.0
	Primary	83	3.5	336	16.5
	Secondary and higher	38	2.3	120	5.0
Sex of the head of the household	Male	140	7.2	851	55.0
	Female	58	2.3	382	35.5
Household wealth index	Poorest	46	2.4	728	60.6
	Poorer	32	1.1	179	9.0
	Middle	34	1.0	121	5.9
	Richer	33	1.1	108	6.4
	Richest	53	3.8	97	8.6
Number of children	1–2	90	4.3	409	23.7
	3–4	36	2.2	337	23.2
	>=5	72	3.0	487	43.6
Mode of delivery	Vaginal	179	8.5	1213	89.3
	Caesarean section	19	1.0	20	1.2

* Weighted percentage; ** < 0.0001.

**Table 2 ijerph-20-06320-t002:** Factors associated with maternal health care utilisation in the emerging regions of Ethiopia.

Variables	Categories	ANC4+			Institutional Delivery			PNC within 24 h		
		Yes	No	AOR (95%CI)	Yes	No	AOR (95%CI)	Yes	No	AOR (95%CI)
Maternal age at the time of delivery (years)	<20	77	173	1	119	131	1	79	171	1
	20–34	293	703	1.21 (0.64–2.28)	416	580	1.42 (0.95–2.13)	252	744	0.82 (0.49–1.37)
	35–49	47	138	1.07 (0.39–2.98)	67	118	1.16 (0.68–1.96)	48	137	0.68 (0.28–1.63)
Residence	Rural	303	872	1	414	761	1	257	918	1
	Urban	114	142	3.59 (1.37–9.46) *	188	68	3.17 (1.23–8.18) *	122	134	1.53 (0.82–2.87)
Region	Afar	108	279	1	105	282	1	77	310	1
	Somali	30	311	0.26 (0.13–0.54) **	71	270	0.71 (0.36–1.40)	30	311	0.43 (0.24–0.78) *
	Benishangul-Gumuz	187	182	2.00 (1.11–3.60) *	231	138	4.70 (1.95–11.32) **	151	218	2.23 (1.16–4.28) *
	Gambela	92	242	0.41 (0.21–0.79) *	195	139	2.61 (1.20–5.69) *	121	213	1.56 (0.86–2.82)
Marital status	Not married/not living with partner				71	54	1	47	78	1
	Married/living with partner				531	775	0.58 (0.28–1.19)	332	974	1.02 (0.57–1.81)
Highest level of maternal education	No education	187	667	1	239	615	1	147	707	1
	Primary	168	251	2.11 (1.31–3.40) **	238	181	1.56 (0.84–2.90)	146	273	1.67 (0.90–3.09)
	Secondary and higher	62	96	2.91 (1.60–5.31) **	125	33	2.82 (0.41–19.36)	86	72	4.80 (2.66–8.65) **
Sex of the head of the household	Male	305	686	1	413	578	1			
	Female	112	328	0.72 (0.39–1.30)	189	251	1.96 (1.26–3.05) *			
Household wealth index	Poorest	137	637	1	175	599	1	104	670	1
	Poorer	78	133	1.59 (0.76–3.33)	112	99	2.24 (1.34–3.75) *	65	146	1.14 (0.67–1.92)
	Middle	63	92	1.36 (0.65–2.84)	89	66	1.37 (0.67–2.77)	57	98	1.31 (0.69–2.52)
	Richer	64	77	1.31 (0.65–2.64)	99	42	3.42 (1.52–7.68) *	59	82	2.91 (0.71–11.99)
	Richest	75	75	1.02 (0.33–3.18)	127	23	6.65 (3.00–14.73) **	94	56	4.55 (1.78–11.57) *
Number of children	1–2	165	334	1	266	233	1	170	329	1
	3–4	100	273	0.77 (0.42–1.42)	133	240	0.43 (0.22–0.86) *	68	305	0.88 (0.45–1.73)
	>=5	152	407	1.04 (0.59–1.82)	203	356	0.92 (0.60–1.40)	141	418	2.26 (1.28–3.99) *
Mode of delivery	Vaginal							350	1042	1
	Caesarean section							29	10	5.48 (1.16–26.00) *

*p* value: * < 0.05, ** < 0.001; OR = Odds Ratio; aOR = adjusted Odds Ratio; ANC4+ = four or more Antenatal care; PNC = Postnatal care; CI = Confidence Interval.

**Table 3 ijerph-20-06320-t003:** Factors associated with the completion of the continuum of maternity care in the emerging regions of Ethiopia (n: 1431).

Variables	Categories	Completion of the Continuum of Maternity Care			
		Yes	No	OR (95%CI)	aOR (95%CI)
Maternal age at the time of delivery (years)	<20	39	211	1	1
	20–34	134	862	0.57 (0.41–0.81) *	0.70 (0.44–1.12)
	35–49	25	160	0.42 (0.25–0.72) *	0.65 (0.32–1.32)
Residence	Rural	132	1043	1	1
	Urban	66	190	3.30 (1.60–6.81) *	1.80 (0.82–3.91)
Region	Afar	43	344	1	1
	Somali	8	333	0.19 (0.06–0.67) *	0.23 (0.07–0.78) *
	Benishangul-Gumuz	103	266	3.21 (1.93–5.35) **	3.41 (1.65–7.04) *
	Gambela	44	290	1.47 (0.80–2.70)	0.67 (0.37–1.21)
Marital status	Not married/not living with partner	23	102	1	1
	Married/living with partner	175	1131	0.59 (0.31–1.10)	0.77 (0.36–1.64)
Highest level of maternal education	No education	77	777	1	1
	Primary	83	336	3.88 (2.48–6.07) **	1.39 (0.93–2.08)
	Secondary and higher	38	120	8.57 (3.97–18.52) **	2.12 (1.13–4.00) *
Sex of the head of the household	Male	140	851	1	1
	Female	58	382	0.48 (0.28–0.85) *	0.96 (0.45–2.05)
Household wealth index	Poorest	46	728	1	1
	Poorer	32	179	2.94 (1.51–5.72) *	1.20 (0.68–2.13)
	Middle	34	121	4.42 (2.01–9.71) **	1.58 (0.74–3.38)
	Richer	33	108	4.44 (2.03–9.72) **	1.63 (0.76–3.50)
	Richest	53	97	11.24 (5.10–24.77) **	4.55 (2.04–10.15) **
Number of children	1–2	90	409	1	1
	3–4	36	337	0.52 (0.27–1.01)	0.96 (0.46–2.02)
	>=5	72	487	0.38 (0.21–0.69) *	1.26 (0.70–2.28)

*p* value: * < 0.05, ** < 0.001; OR = Odds Ratio; aOR = adjusted Odds Ratio; CI = Confidence Interval.

## Data Availability

The data used in the study can be requested and downloaded from Available online: https://www.dhsprogram.com (accessed on 13/10/2022).

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
