# Peer review of "Completion of the Continuum of Maternity Care in the Emerging Regions of Ethiopia: Analysis of the 2019 Demographic and Health Survey"

_ijerph, 2023, doi:10.3390/ijerph20136320_

Round 1

Reviewer 1 Report

Dear Authors,

I found your research interesting and well designed. However, please consider the following comments in hope that these will improve your paper.

Regarding discussion. It would benefit the paper if could contrast your findings with the relevant literature. 

Also, lines 276-282, you comment on the all quintiles but the middle one. Can you please explain and compare its results with the rest. Same comment could accordingly hold to the rest of the discussion parts. 

Please go through editing the English language. Even from the beginning in the abstract there are quite a few spelling mistakes. Also please review lines 31-35 to rephrase and male it easier to understand.

Author Response

Dear Reviewer,

I hope you are doing well.

Kindly find attached the response to your comments.

Best wishes,

Abdulaziz

Reviewer 2 Report

Dear Authors

This is an interesting paper on a particularly important topic, looking at the continuum of maternity care. Below are comments that are aimed at improving the paper.

Introduction

The author(s) mentioned that "These regions are the least developed regions in the country...", this sentence must be referenced.

Methods

The authors must include the institution where ethics were obtained, including the ethics number. In addition, since some participants were adolescents (ages 15 to 19) and not necessarily "women", the recruitment and consent/assent processes must be discussed. In other words, the ethical consideration of including girls as old as 15 years in the study should be discussed.

Results

The maternal age categories must be explained and justified.

Author Response

(The authors gave the same response as above.)

Reviewer 3 Report

Kindly refer to the following comments:

Introduction:

Kindly include a discussion of maternal care (its definition and services involved).

Causes of maternal mortality and relationships to maternity care, specifically among developing countries. 

Methods:

-Provide the sample size calculation.

Best wishes.

Author Response

(The authors gave the same response as above.)
